# GaN/AlN Multi-Quantum Wells Infrared Detector with Short-Wave Infrared Response at Room Temperature

**DOI:** 10.3390/s22114239

**Published:** 2022-06-02

**Authors:** Fengqiu Jiang, Yuyu Bu

**Affiliations:** Key Laboratory of Wide Band-Gap Semiconductor Materials and Devices, School of Microelectronics, Xidian University, Xi’an 710071, China; 15091529270@163.com

**Keywords:** short-wave infrared, GaN/AlN, ISB, infrared detector, quantum well

## Abstract

GaN-based quantum well infrared detectors can make up for the weakness of GaAs-based quantum well infrared detectors for short-wave infrared detection. In this work, GaN/AlN (1.8 nm/1.8 nm) multi-quantum wells have been epitaxially grown on sapphire substrate using MBE technology. Meanwhile, based on this device structure, the band positions and carrier distributions of a single quantum well are also calculated. At room temperature, the optical response of the device is 58.6 μA/W with a bias voltage of 0.5 V, and the linearity between the optical response and the laser power is R^2^ = 0.99931. This excellent detection performance can promote the research progress of GaN-based quantum well infrared detectors in the short-wave infrared field.

## 1. Introduction

As a novel intersubband (ISB) transition detector, GaN-based quantum well infrared detectors have a very high degree of design freedom [1,2,3]. GaN-based quantum well has large band offset of conduction band, short electron relaxation time, and high phonon energy, which make it another infrared detector with research prospect after GaAs-based quantum well infrared detectors. GaN-based infrared detectors use the subband generated by the quantum confinement effect to achieve infrared detection [4]. The conduction band of GaN-based quantum well is large enough (the GaN/AlN quantum well band is the largest, up to 1.96 eV [5]) that enables it to absorb the near-infrared light where the GaAs quantum well infrared detectors are limited due to the small band gap [6]. However, GaN-based infrared detectors are significantly different from commercially available InGaAs infrared detectors in performance indicators. This is because they have essential difference in photoelectric detection mechanism. InGaAs infrared detectors make use of the transition of carriers from valence to conduction band, while GaN-based infrared detectors make use of the intersubband transition of quantum well bands in conduction band. The current optical fiber communication window is 1.3–1.55 μm, which is very suitable for developing GaN-based infrared detectors with fast response speed and high sensitivity. Suzuki et al. found that the relaxation time of electrons between subband in GaN-based quantum well is very fast, as low as 80 fs, so it is very suitable for high-speed light detector preparing, light modulator, and light switch [7]. In addition, GaN material has LO-phonon energy of 92 meV, which is three times higher than that of GaAs material, while ambient heat is about 26 meV at room temperature [8]; therefore, GaN-based infrared detectors can theoretically work at room temperature. Although GaN material has many advantages in the area of light detector, it is restricted by the epitaxy process. How to grow the ultrathin GaN quantum well layer with the high crystal quantum is the key technology to fabricate GaN-based infrared detectors [9,10,11]. And the spontaneous and piezoelectric polarization effects of GaN material itself also make the quantum well band gap tilt, which increase design difficulty of the GaN quantum well [12,13]. In 2003, with the successful preparation of the first GaN-based quantum well ISB infrared detector [14], researchers made many efforts in this field. Subsequent work on short-wave infrared and near-infrared quantum cascade detectors (QCD) is published [15,16,17,18]. Due to the super-wide detection spectrum of GaN-based quantum well infrared detectors, the work of GaN-based quantum well infrared detectors has also been extended to the middle infrared [19,20] and THz band [21,22]. Meanwhile, the simple multi-quantum wells stacked GaN-based quantum well infrared detectors were also published [3,23,24]. In 2018, different from the previous work, Dror et al. epitaxially grew a mid-infrared GaN/AlGaN quantum well cascade detector with well lattice on a silicon (Si) substrate [25]; so far, the work of GaN-based quantum well infrared detector has become more and more abundant. However, how to obtain a high quality GaN/AlN quantum well is still a challenge, such as the experiments on room temperature negative differential resistance [26,27] and the influence of roughness scattering [28] in GaN/AlN interface, which are a great influence on the electrical performance of GaN/AlN quantum well. Therefore, the preparation and development of GaN-based infrared detectors are both opportunities and challenges. At present, the related research work mainly focuses on the preparation of the device, and measurement of the light response performance of the device at low temperature. Thus, how to prepare a GaN-based quantum well ISB infrared detector that can work stably (output stable photoinduced I-V, I-T and other photoelectric properties) at room temperature with excellent high-frequency response character is of great significance.

## 2. Experiment

### 2.1. Device Structure Design and Implementation

In this paper, we study a GaN/AlN multi-quantum wells infrared detector. Figure 1 shows the schematic diagram and the physical drawing of the device, respectively. GaN/AlN quantum wells were successfully grown by MBE (SVTA 35V-2, Eden Prairie, MN, USA) on c-plane of sapphire substrate. First, When the substrate temperature is 810 °C, a thin nucleation layer of AlN is grown on the prepared sapphire substrate. Then the substrate temperature is set to 720 °C and a 700 nm Al_0.6_Ga_0.4_N buffer layer with high Al component is grown on the nucleation layer, which simultaneously acts as the buffer layer to reduce the growth stress of the material and the bottom electrode contact layer. In order to reduce the resistance of the device and increase its conductivity, N-type doping (Si = 5 × 10^19^ cm^−3^) is required. As shown in Figure 2, the XRD peak of Al_0.6_Ga_0.4_N (0002) is 35.70°, indicating that the substrate is of good quality and lays a good foundation for the growth of the following materials. Then, The GaN/AlN quantum wells are grown for 30 cycles. The thickness of the GaN well layers and the AlN barrier layers are both 1.8 nm. By doping the GaN well layers with Si concentration of 1 × 10^15^ cm^−3^, the concentration of free carriers in the quantum well is increased, which enhances the light absorption and light response intensity of the quantum well. Finally, a contact layer of Al_0.6_GA_0.4_N is fabricated with a thickness of 150 nm. Similarly, in order to increase its contact, n-type doping Si = 5 × 10^19^ cm^−3^ is carried out. The position and thickness of GaN/AlN quantum wells are determined by scanning transmission electron microscopy (STEM) (Thermo Fisher, Talos-F200X, Shanghai, China). The doping concentration of Si atom in the contact layer Al_0.6_Ga_0.4_N is obtained by Hall test (As shown in Table 1), and its concentration is very close to 5 × 10^19^ cm^−3^, which meets the design requirements. Due to the layered structure of the device, the actual concentration of carriers in the top contact layer of the device and the quantum well cannot be accurately measured, and can only refer to the design parameters. The active region mesa is 300 μm × 300 μm. Then, electrodes are grown on the top layer and the etched bottom contact layer to complete the device. In order to obtain ohmic contact, Ti/Al/Ni/Au metal electrodes with thickness of 20/160/55/45 nm are grown by magnetron sputter, and they are annealed at 860 °C for 60 s by RTP annealing furnace to increase the electrode contact. The underlying electrode is placed around the active region table to increase the conductivity of the electrode, as shown in the Figure 1b. In order to avoid the influence of the top electrode on the light absorption of the device, the electrode size of 290 μm × 100 μm is prepared on the side of the active mesa of the device.

### 2.2. Structure Characterization and Infrared Detection Methods

The electrical properties of the prepared devices are measured at room temperature using a probe station and a Keithley 4200 semiconductor parameter analyzer (Keithley, 4200A-scs, Shanghai, China). The optical properties of the device are measured by a laser with a wavelength of 1460 nm. We built the test platform by combining a probe station, a Keithley 4200 semiconductor parameter analyzer and an optical power stable 1460 nm laser. The two probes of the probe station are connected to the two electrodes of the device and then connected to the Keithley 4200 semiconductor parameter analyzer. The optical fiber outlet of the laser is fixed 5 cm away from the device. Due to the polarization selection rule of ISB quantum well, a 45° angle between the laser beam and the device plane is required to increase the coupling of the device and light [29]. The bias voltage is added to the device to test the I-T curve of the device under open and closed light. The specific test parameters are given in the Section 3.

### 2.3. Theoretical Calculation Method

We use the self-consistent calculation of Poisson and Schrodinger equation to calculate the band distribution of the quantum well [30,31,32]. Since the quantum well is isotropic in the XY plane where is no quantum limitation of carriers, only the one-dimensional Schrodinger equation in the Z direction, which is the growth direction of the quantum wells, needs to be considered, as shown in Formula (1).
(1)$$-\frac{\hbar^{2}}{2}\frac{d}{dz}\left[\frac{1}{m^{*}\left(z\right)}\frac{d}{dz}\varphi \left(z\right)\right]+V\left(z\right)\varphi \left(z\right)=E\varphi \left(z\right)\;$$
where *ℏ* is reduce Planck’s constant; $m^{*}\left(z\right)$ is the effective mass of the electron; V(z) is the tilted potential energy under the influence of the polarization field; and φ(z) and E are electron wave function and energy eigenvalue respectively. Using the difference method to solve the Schrodinger differential equation:(2)$$\;\frac{df}{dz}\approx \frac{\Delta f}{\Delta z}=\frac{f\left(z+\delta z\right)-f\left(z-\delta z\right)}{2\delta z}\;$$
where δz is the spatial step. The smaller δz is, the closer the difference is to the differential, and the more accurate the calculation results will be. The band structure V(z) of the quantum well and the two-dimensional electron concentration $n_{2D}$ and three-dimensional electron concentration n(z) of the quantum well energy level are obtained by the difference method. Poisson equation is also solved by difference method as follows:(3)$$d/dz[\varepsilon_{0}\varepsilon_{r}\frac{d}{dz}\phi_{H}(z)]=-e[N_{d}{}^{+}(z)-n_{2D}(z)]$$

The electrostatic potential $\phi_{H}\left(z\right)\;$ is generated by the coulomb interaction between charged ions and electrons, where $\varepsilon_{0}$ is the dielectric constant of vacuum and $\varepsilon_{r}$ is the relative dielectric constant. $N_{d}{}^{+}\left(z\right)$ is the donor concentration of ionization, and $n_{2D}\left(z\right)$ is the sum of the concentrations of two-dimensional electrons at all subbands in the quantum well. The correlation potential $\;V_{xc}\left(z\right)$ generated by multi-electrons interaction can be obtained by using the following formula [33]:(4)  Vxc(z)=−0.985e24πε0εr n13(z)×{1+0.034 a*Hn13(z)ln[1+18.376 a*Hn13(z)]}
where e is the electron charge, n(z) is the concentration of the three-dimensional electron in the quantum well, and $a^{*}{}_{H}=4\pi \varepsilon_{0}\varepsilon_{r}\hbar^{2}/m^{*}\left(z\right)e^{2}$ is the effective Bohr radius. The two potential energies generated by electrons $\phi_{H}\left(z\right)$ and $\;V_{xc}\left(z\right)$ are introduced into the conduction potential energy of the quantum well,$\;V\left(z\right)-e\phi_{H}\left(z\right)+V_{xc}\left(z\right)$ substituting the original V(z) into Equation (1). Until the effect of $\phi_{H}\left(z\right)$ and $\;V_{xc}\left(z\right)$ on V(z) is negligible, the iteration ends. The above is the main content of self-consistent calculation of Poisson and Schrodinger equations.

## 3. Results and Discussion

Figure 3a shows the STEM image of GaN/AlN quantum wells in high-angle ring dark field, in which GaN/AlN quantum wells presents a high growth quality. GaN and AlN layers are evenly distributed and clearly layered, which have been marked that the actual thickness of GaN and AlN is 1.8 nm. It can be seen that the quality of the lower quantum wells are deficient, which indicates that there is certain stress between the GaN/AlN quantum well layer and the Al_0.6_Ga_0.4_N layer. According to Figure 3b, we obtained the distribution mappings of Ga, Al, and N elements in the quantum well layer corresponding to Figure 3c–e, respectively. It can be seen that Ga and Al elements have obvious stratification and periodic distribution, while N element presents uniform distribution, indicating that the growth quality of GaN/AlN quantum wells can be guaranteed. However, a small amount of Ga and Al elements are blended into each other; as a result, the conduction band order of the two layers will decreases, corresponding to the reduced subbands spacing in the quantum well; thus, the actual absorption peak wavelength of the quantum well will red shift.

As shown in Figure 4, the theoretical band structure and corresponding absorption wavelength of GaN/AlN (1.8 nm/1.8 nm) quantum well are obtained. Since the device includes the periodic GaN/AlN quantum well with 30 periods, we only need to consider the band position and carrier distribution of a quantum well during the calculation. In Figure 4a, the distribution of the quantum well conduction band formed by GaN and AlN with the difference of conduction band order is calculated. In the quantum well, the position of ground state E0 is 0.25 eV (green straight line), the position of first excited state E1 is 1.118 eV (red straight line), and the energy level difference is 0.868 eV, corresponding to the short-wave infrared wavelength of 1428.6 nm. We use the normalized wave function square to represent the probability of carrier distribution at different positions on the subband. The carrier distribution of the ground state level E0 is represented by the green curve, and the carrier distribution of the first excited state level E1 is represented by the red curve. Figure 4b shows the simulation calculation results of quantum well absorption coefficient of infrared light in the range of 1000 to 2000 nm. The absorption peak of its wavelength is located at 1428.6 nm. This value is the theoretical absorption peak of the quantum well, but in the actual device growth process, due to the complexity of GaN/AlN quantum well growth process, the quality of the quantum well can not be strictly guaranteed, so the quantum well absorption summit is offset on the basis of theory. Therefore, in this work, we use an infrared laser with a wavelength of 1460 nm, which is larger than and close to the theory absorption wavelength of the quantum well, to test the optical response of the device. 

The electrical and optical properties of the prepared devices are measured. The test results are shown in Figure 5a. At room temperature, we test the light response current of the device at different laser powers, since the device is measured at room temperature, inducing a high background noise of the device. Furthermore, the band characteristics of ISB devices also determine the large dark current. Therefore, when the laser power is 30 mW, the obvious light response can be observed. The maximum power that the laser can achieve is150 mW, so we set the power of the laser as 30, 60, 90, 120, and 150 mW.

The I-T curve of the device is measured by Keithley 4200 semiconductor parameter analyzer. Figure 5a shows that the I-V curve of the device is measured under closed light. Then, when the laser power is 150 mW, the light response of the device at different biases is measured. In order to determine the best bias voltage of the device during the test, we calculated the ratio of photocurrent to dark current under different bias voltages of the device, as shown in Figure 5b; when the bias voltage is 0.5 V, the ratio of photocurrent to dark current of the device is the maximum, so we set the test bias of the device as 0.5 V. The top electrode of the device is applied with +0.5 V voltage, and the device current is measured for 10 S with closed light to stabilize the dark current. Then, the laser is turned on to measure the photocurrent of the device for 10 S. Finally, the laser is turned off for 10 S to record the current change of the device. The device current can be obtained by a few μA level changes. In order to study the linearity of the light response of the device with the change of laser power, the light response current of the device at 15 s was intercepted as the light response of the device at the corresponding power. The inset shows the difference of the initial dark current at different powers, the initial dark current at 150 mW is the reference point, as shown in Figure 5c. In order to eliminate man-made errors in the measurement, we test the same device for three consecutive times, take the average value of the photocurrent at each power, make error bar, and perform linear fitting; the results are shown in Figure 5d. The red line in the figure is a fitting line with a linearity of R^2^ = 0.99931, indicating that the optical response of the device presents a good linearity with the increase of the laser power. At this time, the light response of the device can be calculated as 58.6 μA/W at room temperature. The photocurrent is generated by the transition of electrons in the quantum well. This process is the transition of electrons absorbing infrared light from the ground state to the first excited state. Therefore, the photocurrent presents a good linearity with the power of the laser.

## 4. Conclusions

Finally, a GaN/AlN multi-quantum wells infrared detector with a response wavelength of 1460 nm is designed, epitaxially grown, and fabricated. We obtain the theoretical absorption peak of GaN/AlN quantum well by self-consistent calculation of Poisson–Schrodinger equation. At room temperature, the optical response of the device is 58.6 μA/W, and the intensity of the optical response has a excellent linearity with the laser power change. In this work, the successful preparation of GaN/AlN multi-quantum wells infrared detector has enriched the relevant work and provided a feasible reference for the subsequent research work.

## Figures and Tables

**Figure 1 sensors-22-04239-f001:**
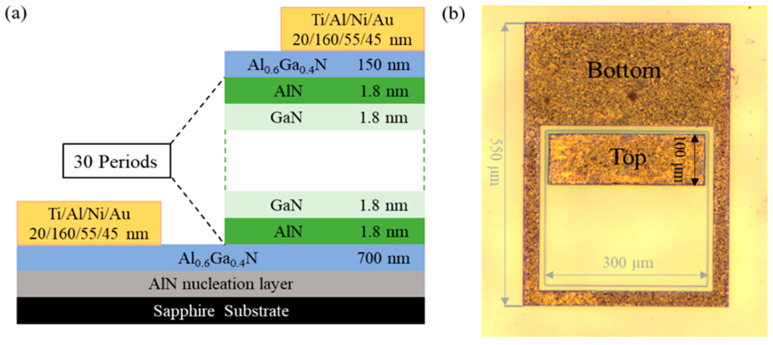
(**a**) Structure diagram of the GaN/AlN multi-quantum wells infrared detector. The omitted part is 30-periods GaN/AlN quantum wells starting and ending with the AlN layer. The Si atom doping concentration of GaN wells layer is 1 × 10^15^ cm^−3^, and the Si atom doping concentration of top and bottom Al_0.6_Ga_0.4_N contact layer is 5 × 10^19^ cm^−3^. (**b**) Physical picture of the GaN/AlN multi-quantum wells infrared detector. The outer layer is the bottom electrode of the device, and the middle is the top electrode of the device.

**Figure 2 sensors-22-04239-f002:**
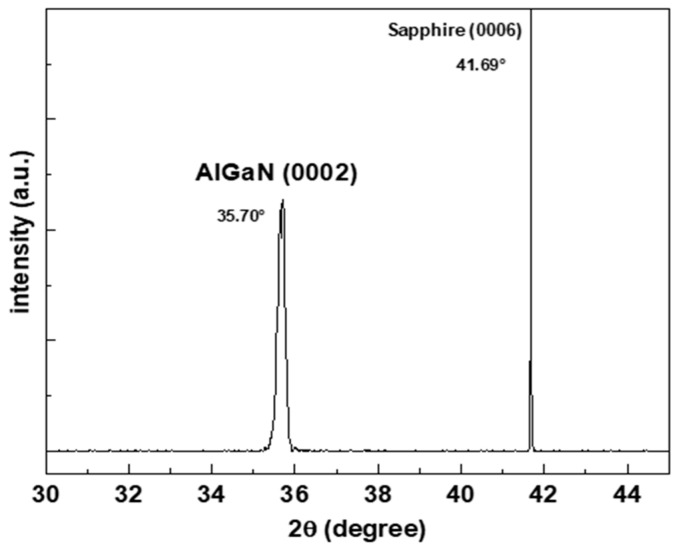
On sapphire substrate, the Al_0.6_Ga_0.4_N buffer layer was characterized by XRD.

**Figure 3 sensors-22-04239-f003:**
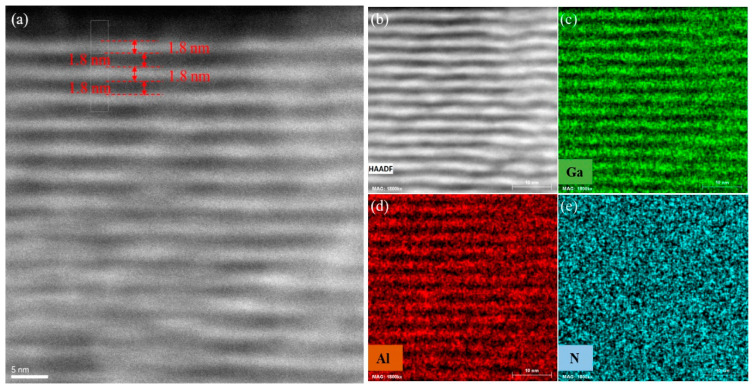
(**a**) GaN/AlN quantum wells scanning transmission electron microscope high-angle annular dark-field image (STEM HAADF). Due to higher Z contrast, the brightness of GaN layer is higher than that of AlN. It is marked in the figure that the thickness of GaN and AlN are 1.8 nm. (**b**) Scanning transmission electron microscope high-angle annular dark field image(STEM HAADF) of GaN/AlN quantum well with 10 nm scale. Used to measure the distribution of elements in the quantum well layers. (**c**–**e**) are, respectively, the distribution mappings of Ga, Al, and N elements in the corresponding quantum well layers of (**b**).

**Figure 4 sensors-22-04239-f004:**
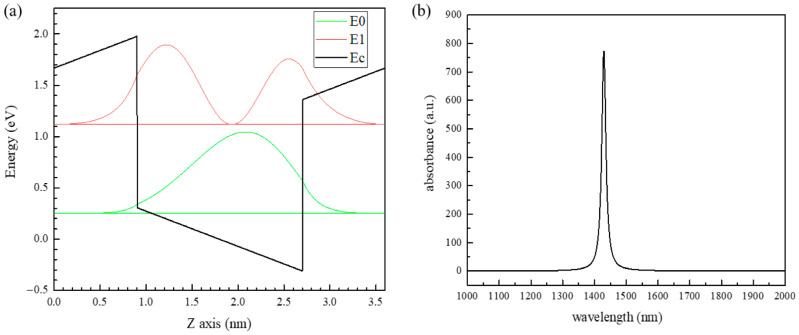
(**a**) Band position and carrier distribution of the one-period GaN/AlN (1.8 nm/1.8 nm) quantum well. The black line is the conduction band of the quantum well, the conduction band of AlN on both sides, and the conduction band of GaN in the middle. The green lines are the position of the ground state level E0 in the quantum well (straight line) and the distribution of carriers in the ground state (curved line). The red lines show the position of E1 in the first excited state in the quantum well (straight line) and the distribution of carriers in the first excited state (curved line). (**b**) The simulation results of absorption coefficient of infrared light in the range of 1000 to 2000 nm for the GaN/AlN (1.8 nm/1.8 nm) quantum well.

**Figure 5 sensors-22-04239-f005:**
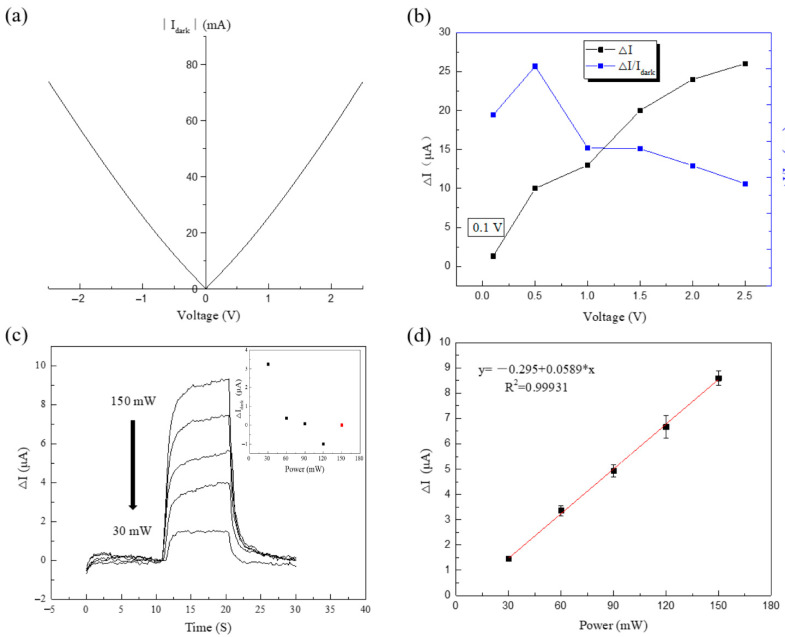
(**a**) The absolute value of dark current(∣I_dark_∣) is measured at different bias voltage. (**b**) When the laser power is 150 mW, the light response of the device at different biases is measured (black line), and the ratio of the photocurrent to the dark current is get (blue line). (**c**) I-T curves of the light response of the device when the light is on and off at different laser powers. The inset shows the difference of the initial dark current at different powers, with the initial dark current at 150mW (red dot) as the reference point. (**d**) At 15 s, the photocurrent of the device at different laser powers.The red line is the fitting line of the photocurrent of the device with the change of laser power. The black dots represent the average of the results, and the short lines represent the standard deviation.

**Table 1 sensors-22-04239-t001:** Hall test results of the Al_0.6_Ga_0.4_N buffer layer.

Parameter (Unit)	Value
The type of semiconductor	N-type
Resistivity (ohm·cm)	4.2475 × 10^−3^
Hall coefficient (cm^3^/C)	−1.3352 × 10^−1^
Hall mobility (cm^2^/(v·s)	3.1436 × 10^−1^
Carrier concentration (cm^−3^)	4.6749 × 10^19^

## Data Availability

The data presented in this study is available on request from the corresponding author.

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
