# Peer review of "GaN/AlN Multi-Quantum Wells Infrared Detector with Short-Wave Infrared Response at Room Temperature"

_sensors, 2022, doi:10.3390/s22114239_

Round 1

Reviewer 1 Report

The authors have fabricated and characterized an infrared photodetector based on intersubband transitions in GaN/AlN quantum wells. In my opinion, the manuscript is suitable for the Sensors MDPI journal, and can be accepted after some improvements:
1) Authors have achieved the device responsivity of 62 μA/W at wavelength of 1460 nm, while commercially available InGaAs and Ge photodiodes have responsivity of about four orders of magnitude higher. It should be discussed in the text how the obtained parameters are related to the state of the art in this field, including III-Nitride photodetectors and other material systems. If the device parameters have been improved, the authors should emphasize which changes in technology or structure design have resulted in the improvements.
2) It would be useful to add more details about MBE growth process, for example growth temperatures and type of the nitrogen source, and some parameters characterizing the crystal quality of the structure, for example dislocation density.
3) It would be useful to show the experimental spectrum of the photocurrent or the absorption spectrum to see how close is the measured point to the maximum. In the conclusion "response wavelength of 1460 nm" can be uderstood as the peak wavelength of the photocurrent, but this is the wavelength at which the photocurrent was measured.  
4) With what accuracy was the laser power measured and whether it was a rectangular function in time?  What is the cause of the transient processes observed in the figure 4a? Are they associated with the laser power profile or with the response time of the photodetector?
5) In the figure 4a dark current is subtracted, it would be useful to add its value in the text. Why the linear fit in the figure 4b has a negative offset? I think it would be better to use the maximum absolute nonlinearity with estimation of its accuracy for characterization of the photodetector linearity instead of the coefficient of determination R2, which was obtained without taking into account the experimental accuracies. Also it would be useful to add the error bars for the photocurrent and for the laser power in the figure 4b.    
6) At the page 2 the concentration obtained by Hall test is free electron concentration but not the Si atom concentration. In the manuscript text the Hall test results are reported for the AlGaN layer, but in the caption of the table 1 for the "GaN/AlN multi-quantum wells".
7) Equation 1-1 should be with h_bar Planck constant.
8) The authors should add the reference for the used parametrization of the exchange-correlation potential (equation 1-4).
9) At the page 5 "laser power is 30 mV"  - replace "mV" by "mW".
10) In the abstract "can supply the inadequacy"  - may be need to rephrase.
                                "have been epitaxial"        - epitaxially grown?
11) At the page 1 "large band order"      -  band offset?
                               "high crystal quantum"  -  quality?
    At the page 2 "challenge Therefore"   -  "challenge. Therefore"
    At the page 6 "designed, epitaxial and fabricated" - epitaxially grown?

Author Response

To Reviewer #1:

  1. Authors have achieved the device responsivity of 62 μA/W at wavelength of 1460 nm, while commercially available InGaAs and Ge photodiodes have responsivity of about four orders of magnitude higher. It should be discussed in the text how the obtained parameters are related to the state of the art in this field, including III-Nitride photodetectors and other material systems. If the device parameters have been improved, the authors should emphasize which changes in technology or structure design have resulted in the improvements.

Response: Thanks for the constructive suggestion. Following the suggestion, the introduction section of this manuscript has been modified. We stated that the GaN-based infrared detectors and InGaAs infrared detectors have essential difference in photoelectric detection mechanism. Therefore,The performance indicators of them are significantly different.

  1. It would be useful to add more details about MBE growth process, for example growth temperatures and type of the nitrogen source, and some parameters characterizing the crystal quality of the structure, for example dislocation density.

Response: Thanks a lot for your constructive suggestion. We have added the temperature parameters of the substrate during material growth and XRD patterns that characterize the Al0.6Ga0.4N lattice quality.

  1. It would be useful to show the experimental spectrum of the photocurrent or the absorption spectrum to see how close is the measured point to the maximum. In the conclusion "response wavelength of 1460 nm" can be uderstood as the peak wavelength of the photocurrent, but this is the wavelength at which the photocurrent was measured.

Response: Thanks for your kind suggestions. Because the device is tested at room temperature, the noise is very large. For the photocurrent to be observed, the light intensity must be at least a few mW. while the continuously variable single-wavelength infrared source cannot provide such a large light intensity, so it is difficult to obtain the photocurrent spectrum of different wavelengths in the experiment. In the test of Fourier infrared absorption spectrum, the material growth defect was troubled, and the position of absorption peak could not be determined. We can only determine the approximate range of absorption peak through literature reference, theoretical calculation and specific sample analysis. We can only confirm that the wavelength of 1460nm is near the absorption peak and the device can respond to it, but it is not the absorption wavelength of maximum response.

  1. With what accuracy was the laser power measured and whether it was a rectangular function in time? What is the cause of the transient processes observed in the figure 4a? Are they associated with the laser power profile or with the response time of the photodetector?

Response: Thanks for your kind question. During the measurement, the output of the laser is not a rectangular function in time, and there is a certain delay when the laser is on and off that is considered in the waveforms of the rising and falling edges of the device photocurrent.

  1. In the figure 4a dark current is subtracted, it would be useful to add its value in the text. Why the linear fit in the figure 4b has a negative offset? I think it would be better to use the maximum absolute nonlinearity with estimation of its accuracy for characterization of the photodetector linearity instead of the coefficient of determination R2, which was obtained without considering the experimental accuracies. Also it would be useful to add the error bars for the photocurrent and for the laser power in the figure 4b.

Response: Thanks for the constructive suggestion. We supplement the dark current I-V test curve of the device and give the difference of the initial dark current measured at different powers. Because the device is tested at room temperature, the noise is very large. For the photocurrent to be observed, the light intensity must be at least a few mW. Therefore, the fitting curve intersects the X axis (Power axis) in a positive direction, resulting in a negative offset. At the same time, in order to avoid the error of the value, we use the device results with three tests to make error bar. The new data processing method reduces the negative bias of the fitting curve, improves the linearity of the light response of the detector, and makes the experimental results more reliable. 

  1. At the page 2 the concentration obtained by Hall test is free electron concentration but not the Si atom concentration. In the manuscript text the Hall test results are reported for the AlGaN layer, but in the caption of the table 1 for the "GaN/AlN multi-quantum wells".

Response: Thanks for the constructive suggestion. We have standard “GaN/AlN multi-quantum wells” as “Al0.6Ga0.4N buffer layer”. in this article. And the changes in the text have been highlighted by red color in the revised manuscript compared with that of the previous version.

  1. Equation 1-1 should be with h_bar Planck constant.

Response: Thanks for the constructive suggestion. We have standard “h” as “h_bar”. And the changes in the text have been highlighted by red color in the revised manuscript compared with that of the previous version.

  1. The authors should add the reference for the used parametrization of the exchange-correlation potential (equation 1-4).

Response: Thanks for the constructive suggestion. The references to the formula have been added.

  1. At the page 5 "laser power is 30 mV"  - replace "mV" by "mW".

Response: Thanks for the constructive suggestion. We have corrected this question in the article and marked it in red.

  1. In the abstract "can supply the inadequacy"  - may be need to rephrase.

                                "have been epitaxial"        - epitaxially grown?

Response: Thanks for your kind question. We have corrected these questions in the article and marked them in red.

  1. At the page 1 "large band order"      -  band offset?

                               "high crystal quantum"  -  quality?

    At the page 2 "challenge Therefore"   -  "challenge. Therefore"

    At the page 6 "designed, epitaxial and fabricated" - epitaxially grown?

Response: Thanks a lot for your constructive suggestion. We have corrected these questions in the article and marked them in red.

Reviewer 2 Report

The authors reported their results on MBE grown AlN/GaN multi-quantum wells as infrared detector operating at room temperature. The detector shows reasonable optical response at 1460 nm laser with good linearity. However, before publication, some minor revisions are required:

(1)   The authors reported the optical response at 1460 nm laser. It would help the readers if the response at different wavelengths can be measured and compared with calculation. Or the authors could comment on why this special wavelength is selected.

(2)   Like in (1), only the response at 0.5 V is reported. What do the dark current and photocurrent look like at different voltages? 

(3)   As mentioned by the authors in page 2, high quality GaN/AlN growth is concluded as the main challenge. This is too general. It’s better to describe the challenges and related investigations more clearly, such as the critical effect of dislocations and scattering from interface roughness and alloy disorder or phonons, etc., which have been studied in some recent papers (https://doi.org/10.1002/adfm.202007216; https://doi.org/10.1186/s11671-019-3043-6; https://doi.org/10.1002/aelm.201800651;).

Author Response

To Reviewer #2:

  1. The authors reported the optical response at 1460 nm laser. It would help the readers if the response at different wavelengths can be measured and compared with calculation. Or the authors could comment on why this special wavelength is selected.

Response: Thanks for your kind suggestions. Because the device is tested at room temperature, the noise is very large. For the photocurrent to be observed, the light intensity must be at least a few mW. while the continuously variable single-wavelength infrared source cannot provide such a large light intensity, so it is difficult to obtain the photocurrent spectrum of different wavelengths in the experiment. We can only determine the approximate range of absorption peak through literature reference, theoretical calculation and specific sample analysis. We can confirm that the wavelength of 1460nm is near the absorption peak and the device can respond to it.

  1. Like in (1), only the response at 0.5 V is reported. What do the dark current and photocurrent look like at different voltages? 

Response: Thanks for the constructive suggestion. We supplement the dark current I-V test curve of the device and give the difference of the initial dark current measured at different powers. When the laser power is 150 mW, the light response of the device at different biases is measured, as shown in Figure 5 (b).

  1. As mentioned by the authors in page 2, high quality GaN/AlN growth is concluded as the main challenge. This is too general. It’s better to describe the challenges and related investigations more clearly, such as the critical effect of dislocations and scattering from interface roughness and alloy disorder or phonons, etc., which have been studied in some recent papers. (https://doi.org/10.1002/adfm.202007216;

https://doi.org/10.1186/s11671-019-3043-6; https://doi.org/10.1002/aelm.201800651;).

Response: Thanks for the constructive suggestion. We have described the challenges and related investigations more clearly and added the references.